# MFCL Vision: Benchmarking Tool Use in Multimodal Large Language Models for Visual Reasoning Tasks

**Huanzhi Mao**[*]
University of California, Berkeley
huanzhimao@berkeley.edu

**Jad Bendarkawi**[*]
Princeton University
jadb@alumni.princeton.edu

**Evan Turner**[*]
University of Maryland, Baltimore County
eturner1@umbc.edu

**Ritesh Chavan**[*]
Stony Brook University
riteshsunil.chavan@stonybrook.edu

## Abstract

As multimodal large language models become tool-using agents, the field still lacks a standardized metric for translating visual inputs into correct tool invocations. We introduce **MFCL Vision**, the first large-scale benchmark for *vision-based function calling*, comprising **250** expert-verified tasks across five image domains (*Places*, *Events*, *Media*, *Sports*, *Shopping*) and five query types (*Locate*, *Temporal*, *Select*, *Identify*, *Quantify*). Each task comprises (1) a textual user query, (2) an accompanying image, (3) a ground-truth answer obtained from the web, and (4) a human-produced reasoning trace for comparative error analysis. To constrain the task, we expose a singular web-search tool to each model. To examine the robustness of multimodal LLMs' perception-to-tool-use pipeline, we introduce controlled visual perturbations, including crops, resizes, and color channel removal. Our automatic grader computes exact-match scores on model final answers, removing dependence on brittle and potentially biased LLM judges. We evaluate leading models and present a taxonomy of failure modes, including visual reasoning, keyword selection, and tool avoidance errors. By releasing MFCL Vision's dataset, taxonomy, and diagnostics, we aim to accelerate progress towards versatile multimodal agents capable of intelligent tool usage in complex visual contexts.

## 1 Introduction

Recent advances in multimodal LLMs have demonstrated their potential in agentic, long-horizon use cases. However, effective real-world deployment necessitates the ability to synthesize and leverage visual and textual information in order to reason about the world and autonomously pursue goals. Increasingly, use cases require agents to interface with external tools (e.g. robotic controllers, database lookups, API endpoints, etc.) to either retrieve up-to-date information or enact actions that have real-world consequences. This paradigm, known as *function calling*, has emerged as the fundamental driver in transitioning LLMs from reasoning engines into actionable agents.

Despite the commercial excitement surrounding AI agents, robust *vison-based* function calling remains an open challenge. This perception-to-tool-use capability is especially pertinent for models deployed in accuracy-critical domains like analyzing medical imagery or interpreting enterprise data visualizations. Also, standing perception problems in LLMs are further exacerbated by a lack of systematic evaluation for how effectively existing models execute vision-based function calling end-to-end . Current benchmarks evaluate either (i) text-only tool use—e.g., BFCL [Patil et al.,

---

[*]Equal Contribution.

Submitted to 39th Conference on Neural Information Processing Systems (NeurIPS 2025). Do not distribute.

2025], T-Eval [Chen et al., 2024], and $\tau$-BENCH [Yao et al., 2024]—or (ii) general multi-modal understanding such as MMMU [Yue et al., 2024]. These frameworks overlook real-world settings in which tools must be utilized deliberately and productively despite sparse visual information.

We introduce **MFCL Vision** (**M**ultimodal **F**unction **C**alling Eva**L**uation – Vision Suite), the first benchmark to address this gap. MFCL Vision is a framework containing *250* expert-verified vision-based function calling tasks. Each task specifies (1) a user query, (2) a contextualizing image, (3) a ground-truth answer and (4) a human-produced reasoning trace to validate correct application of the tool. The human-produced reasoning traces are used to manually analyze model traces against a validated example to identify root causes for task failure. Our annotation protocol yields **exact-match** references, enabling automatic grading without relying on fragile or biased LLM judges.

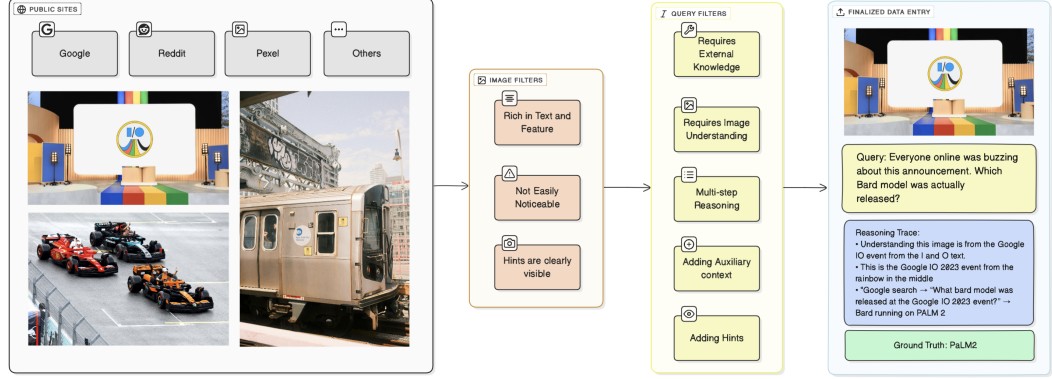

Figure 1: Data construction pipeline for MFCL Vision. We curate both user-contributed and public-source images (Google, Reddit, Pexels, etc), filter for salient visual cues, and write queries requiring external knowledge. We optionally tune queries to enforce multi-step visual reasoning or provide additional context or hints.

Our analysis uncovers three overarching failure modes: (i) **Avoiding Tool Use**, in which models make an unsubstantiated guess, abstain, or ask follow-up questions; (ii) **Poor Keyword Selection**, where generated queries are overly simplistic, vague or irrelevant; and (iii) **Visual Reasoning Errors**, which range from misreading text to misinterpreting spatial elements. We further present auxiliary failure modes that fall within these three broader error categories. To disentangle perception from reasoning, and to study the effectiveness of different image-training paradigms, we run controlled ablations on each image, including grayscale conversions, canny-edge filters, color jittering, and partial occlusion. Through our study, we formulate and present the first error taxonomy for vision-based function calling to guide future research towards robust and versatile multimodal AI agents.

In summary, this work makes the following contributions:

1. We propose **MFCL Vision**, the first benchmark to systematically evaluate vision-based function calling in multimodal LLMs under real-world visual constraints.

2. We curate and release *250* image-query tasks with ground-truths and automated grading.

3. We conduct a large-scale study of models and identify and analyze dominant failure modes, providing actionable insights to accelerate progress toward reliable multimodal AI agents.

## 2 Related Work

### 2.1 Tool Calling Benchmarks

Given the growing recognition of tool-calling capabilities as crucial in ensuring LLM agents generalize across application domains outside of text, various works have proposed evaluation techniques and targeted benchmarks. Early benchmarks such as TOOLBENCH [Qin et al., 2023], API-Bank [Li et al., 2023], and GORILLA API BENCH [Patil et al., 2023] focus on text-only scenarios, where the model must map a prompt with text-only tokens to a corresponding function-calls in various languages. More recent efforts like BFCL [Patil et al., 2025] and $\tau$-Bench [Yao et al., 2024, Barres

et al., 2025] broaden the scope to include multi-turn and multi-step tool-use with intermediate user interactions. These datasets are limited in scope by pertaining solely to text, which constrains the amount of context available to the model. MFCL Vision builds upon and extends existing tool-calling research by accounting for the vision modality along with perturbations that highlight failures at the boundaries of perception, grounding, and formatting.

## 2.2 Vision-based Tool Calling Benchmarks

Robust and reproducible evaluations for tool use in vision-related tasks remains nascent. A handful of works have explored letting language models plan or execute tools to solve visual problems. For example, VISPROG [Gupta and Kembhavi, 2022] and VIPERGPT [Surís et al., 2023] employ LLMs to generate Python-like code plans that invoke vision models step-by-step, and then execute these plans to answer visual queries. Systems like HUGGINGGPT and VISUAL-CHATGPT similarly use an LLM to orchestrate external models (e.g. image captioning) as tools [Shen et al., 2023, Wu et al., 2023, Huang et al., 2025]. The OSWORLD benchmark [Xie et al., 2024] evaluates how effectively multimodal LLMs operate software environments. Conditioned with user screenshots and filtered a11y trees with coordinates, models were tasked with executing the correct sequence of actions to accomplish a certain task. MFCL Vision diverges from OSWorld's approach in its emphasis on naturalistic, real world images rather than constraining the visual context strictly to computer use.

We build on the previous literature by (1) intentionally designing MFCL Vision tasks to go beyond simple visual reasoning, stressing perception-to-tool-use capabilities under real-world noise, occlusion, and distractors and (2) establishing a reproducible, unified evaluation protocol that quantifies tool-augmented reasoning through consistent metrics applicable across models and tasks.

## 3 The MFCL Vision Dataset

The MFCL Vision dataset comprises 250 image-query tasks spanning five image domains (*Places*, *Events*, *Media*, *Sports*, *Shopping*) and five query types (*Locate*, *Temporal*, *Select*, *Identify*, *Quantify*) (Figure 2). Successful completion of each task requires synthesizing user intent with image information to conduct external web search. For each task, there are a number of valid pathways the model can take to find the correct answer. This open-endedness enables models to demonstrate their agentic capabilities, as it requires them to balance avoidance under uncertainty with confidence in potentially overrepresented search results.

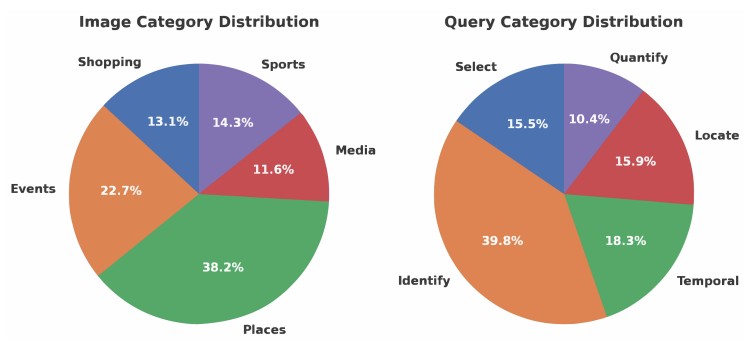

Figure 2: MFCL Vision category distributions. Left: image categories—*Places* and *Events* are most prevalent. Right: query categories—*Identify* dominates, followed by *Temporal* and *Locate*.

## 3.1 Dataset Construction

We create the dataset based on the following principles: **Salient visual hints:** Each image contains visual clues the model can link with details in the user prompt to conduct a promising first search step. By using images with sufficient resolution and naturalistic composition, our benchmark evaluates each model's baseline visual reasoning capabilities. **Dependent on external evidence:** For web search tasks, queries are designed such that the answer cannot be obtained without consulting up-to-date

external sources. Additionally, these do not collapse into pure visual reasoning tasks (e.g., OCR, object recognition, or simple counting). **Dependent on the image context:** The image provides essential disambiguation. Without it, the query is unanswerable (e.g., asking "Who owns this team?" without showing a logo). This ensures that both visual understanding and query specifications must interact to produce productive search queries. **Solvability:** Each task is constructed such that a non-expert human with access to web-search could answer the question correctly.

### 3.2 Entry Tuning and Validation

Each task undergoes iterative tuning (Figure 3) to calibrate difficulty. We noted a trade-off between creating prompts that are not artificially contrived for difficulty yet still require the models to accurately synthesize subtle image and textual details in order to construct web searches. As such, our tuning process includes adding conversational context to emulate user personas, enforcing multi-step search chains, injecting lightweight hints, or selectively cropping the image. We only retain entries that are solvable by humans yet consistently defeat state-of-the-art LLMs. These entries produced phenomena including repeated incorrect answers, excessively long reasoning traces, verbose non-answers, expressions of uncertainty, and outright refusals to attempt a solution.

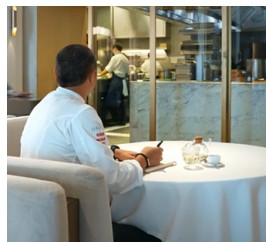

I'm a food critic by trade and I love trying new restaurants while I'm on vacation, especially when I'm traveling abroad. Looking at this photo I took while dining in Southeast Asia, can you tell me which French department the restaurant's head chef was born in? Only return a single string.

Figure 3: Anatomy of an image-query entry that received several tuning treatments. The query is decomposed into distinct components: orange–auxiliary context, blue–the actual information request, green–query hints to constrain the search space, and pink–return format for string matching.

## 4 Evaluation Methodology

The MFCL Vision evaluation pipeline extends the BFCL framework [Patil et al., 2025] by incorporating image inputs into API calls for all integrated models and exposing a singular `search` tool. This design enables plug-and-play compatibility for new models, straightforward integration for the open-source community, and scalable benchmarking for rapid iteration.

**Exact-match accuracy.** Each model receives a system prompt specifying the expected output format (Appendix A.2). We compute exact match on the `answer` field only, after lower-casing and removing punctuation, thereby avoiding false positives from partial matches.

**Ablations.** We conduct three ablations: (i) *color manipulations*—conversion to grayscale or red-green only; (ii) *edge-based transformations*—standard edge detection; and (iii) *aspect-ratio changes*—cropping or resizing to 4:3 or 16:9. Crops preserve key regions, while resizing maintains full content at the cost of possible distortion.

**Large-scale model support.** Models with native function calling (e.g., GPT-5, Gemini-2.5-Pro, Claude-4.1-Opus) execute MFCL Vision directly by receiving all function definitions via the `tools` field. Prompt-only models lack such interfaces and instead generate structured function calls through system-prompt-based emulation, following BFCL [Patil et al., 2025].

## 5 Results and Error Analysis

Our discussion centers on six recurring failure modes that underscore core obstacles must be overcome for LLMs with vision-based function calling to achieve reliable performance in real-world contexts. For Model Performance results, refer to Table 1. See Appendix A.1 for a distribution of prominent error types across a sample of select models.

Table 1: Model performance on MFCL Vision across all 8 variations.

| Model | Overall | Base | Crop16:9 | Crop4:3 | Resize16:9 | Resize4:3 | B&W | Edge | Red&Green |
|---|---|---|---|---|---|---|---|---|---|
| GPT-5-2025-08-07 (FC) | **29.3** | **34.7** | **31.9** | **31.1** | **30.7** | **32.7** | **27.1** | **17.1** | **29.1** |
| Gemini-2.5-Pro (FC) | 26.6 | 29.9 | 31.1 | 28.7 | 25.9 | 29.5 | 25.5 | 14.3 | 27.9 |
| Gemini-2.5-Flash (FC) | 23.1 | 25.5 | 26.7 | 23.1 | 25.5 | 24.3 | 23.1 | 12.4 | 23.9 |
| Grok-4-0709 (FC) | 22.7 | 25.1 | 25.1 | 25.5 | 21.1 | 25.5 | 22.3 | 11.6 | 25.1 |
| o4-mini-2025-04-16 (FC) | 20.0 | 23.1 | 22.7 | 23.5 | 18.7 | 20.7 | 19.9 | 11.2 | 20.3 |
| Claude-Opus-4.1 (FC) | 15.9 | 16.7 | 18.3 | 17.5 | 15.9 | 17.5 | 15.5 | 11.6 | 13.9 |
| Claude-Sonnet-4 (FC) | 14.9 | 16.7 | 17.5 | 17.9 | 13.9 | 18.3 | 12.4 | 8.4 | 14.3 |
| GPT-4o-2024-11-20 (FC) | 11.7 | 12.0 | 14.3 | 15.5 | 10.4 | 12.8 | 12.0 | 4.8 | 11.6 |
| Llama-4-Maverick (FC) | 10.6 | 12.8 | 11.2 | 9.6 | 10.8 | 12.4 | 11.6 | 6.8 | 10.0 |
| Amazon-Nova-Pro-v1:0 (FC) | 10.1 | 12.8 | 10.4 | 9.6 | 9.6 | 10.4 | 9.6 | 6.8 | 11.6 |
| GPT-4o-mini-2024-07-18 (FC) | 9.0 | 10.0 | 11.2 | 8.4 | 10.0 | 9.2 | 9.2 | 6.4 | 8.0 |
| Mistral-Medium-2508 (FC) | 8.7 | 10.4 | 10.8 | 8.4 | 10.8 | 11.2 | 9.2 | 1.2 | 8.0 |
| Pixtral-Large-2411 (Prompt) | 8.4 | 9.6 | 10.0 | 7.6 | 8.8 | 11.2 | 6.0 | 6.0 | 8.0 |
| GLM-4.5V (Prompt) | 7.9 | 10.0 | 5.2 | 9.2 | 7.2 | 10.4 | 8.8 | 3.6 | 8.8 |
| Command-A-Vision-07-2025 (Prompt) | 6.2 | 6.8 | 6.0 | 6.0 | 7.2 | 6.0 | 7.2 | 4.4 | 6.0 |

**FM 1A: Visual Reasoning Errors**  Failures largely stemmed from known weaknesses of VLMs that were deliberately targeted during image curation and tuning. Models often struggle to balance textual with visual information and face challenges with distinct spatial arrangements.

**FM 1B: Subset Confusion**  Faced with many look-alike objects, models often misidentify subsets. In overloaded scenes they tend to omit items or hallucinate extras, skewing keywords (Figure 4).

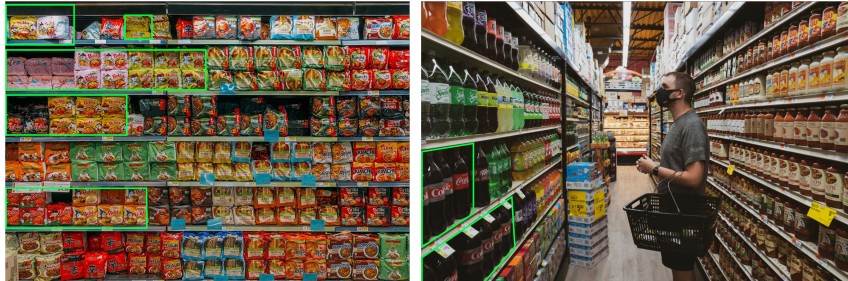

Figure 4: Example images that induced subset confusion. Green boxes mark target subsets, but models often hallucinate extra items or miss targets, especially in the *Shopping* category.

**FM 1C: Myopia**  In images with significant depth of field, models over-attend to salient foreground objects at the expense of more relevant background cues—even when named in the query (Figure 5). This foreground bias skews search, with distractors driving keyword selection.

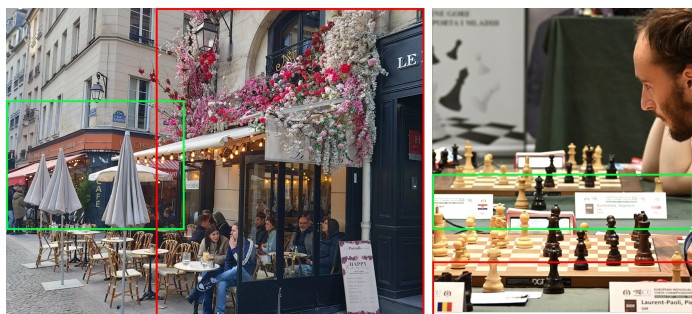

Figure 5: Example images that induced myopic behavior. The green box indicates the key clue and the red box highlights the foreground distractor. Both examples had associated queries that explicitly informed the model to reason on something further back in the image (e.g. referring to the *"cafe with orange signage"* or *"chessboard in the back"*).

**FM 2: Avoiding Tool Use** Sometimes models avoid searching when uncertain. Instead, they either ask clarifying questions (Figure 6) or provide best-guess answers based solely on internal reasoning despite having access to search tools (Figure 7).

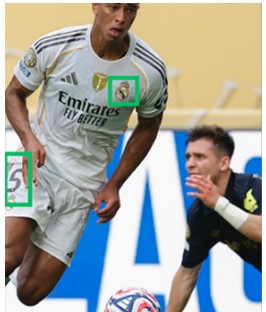

Figure 6: Example of avoiding tools use via refusal. This entry displays the player's team logo and jersey number, which are major clues the models should recognize before making any tool calls. Despite this, the model stalled and requested more input.

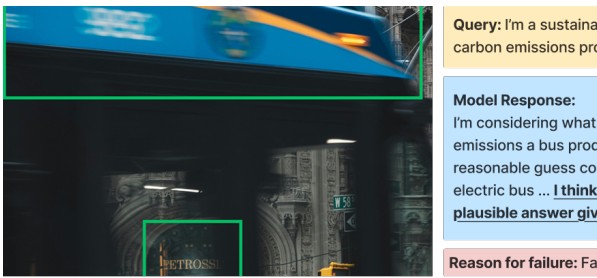

Figure 7: Example of avoiding tool use via search-blind reasoning. Rather than reasoning about the image or leveraging search, the model simply makes a blind guess, offering meta-reasoning about what might be a "plausible" answer given the query text.

**FM 3: Poor Keyword Selection** Models generate vague or irrelevant search queries. Rather than extracting specific visual details (e.g., text, symbols, spatial cues) into targeted keywords, they default to generic phrases (Figure 8), yielding irrelevant search results.

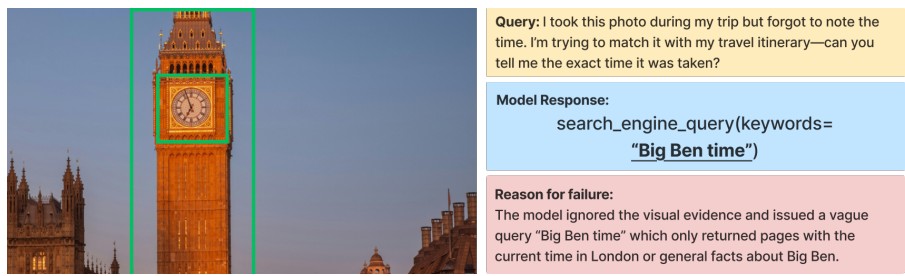

Figure 8: Example of poor keyword selection. This entry implicates analog clock reading and spatial orientation relative to the sun to determine the exact time. The model failed to combine any of this nuance to produce specific keywords for search.

**FM 4: First-hop Bias** Models latch onto an initial assumption and resist updating, even when faced with contradictory evidence (sometimes rationalized away as *"misremembering"*, Figure 9). In other cases, models reason correctly during the trace, but the final output reverts to the initial incorrect guess, discarding their own reasoning.

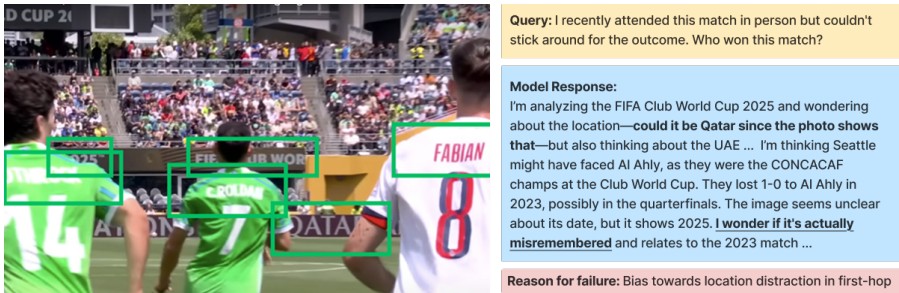

Figure 9: Example of first-hop bias. Even with clear visual evidence of teams, players, and year, the model fixates on a Qatar Airways ad during its first reasoning step. This anchors it to an incorrect path and leads it to dismiss the correct temporal evidence.

**FM 5: Over-reliance on Query Text**   Models prioritize keywords from the provided user query over the information present in the image. Instead of grounding reasoning in visual evidence (e.g., on-screen text, logos, etc.), they issue tool calls based on phrases pulled from the query (Figure 10).

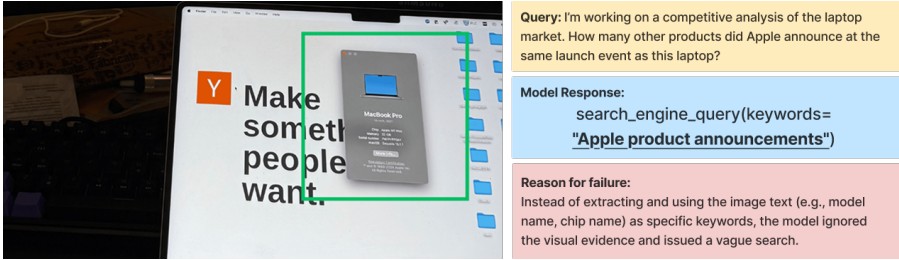

Figure 10: Example of over-reliance on query text. Despite the clear *About This Mac* window, the model ignored the text and issued a vague search based only on the provided query.

**FM 6: Abandoning promising leads**   Models note alternative hypotheses but settle on the first plausible answer, often reinforced by search hits, leaving correct options unexamined (Figure 11).

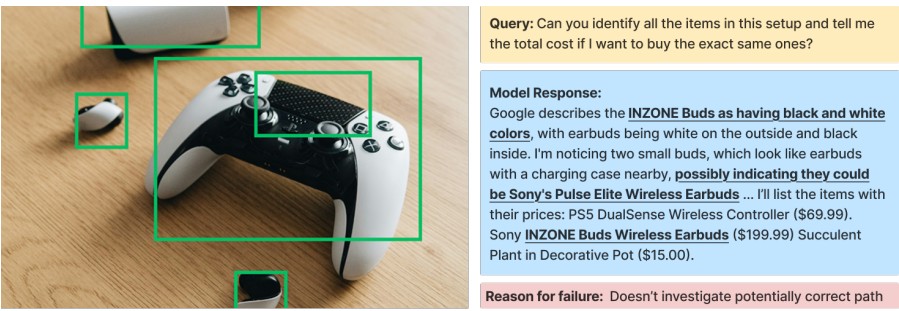

Figure 11: Example of abandoning a promising lead. While able to surface potential answers, the model latches onto the first plausible path and ignores the raised alternative which happened to be correct.

## 6   Ablation Analysis

Examining ablation accuracies (Table 1), the *Edge Detection* variant produces the largest performance drop; all other treatments yield only minor degradation.

**Edge detection removes key cues**   The sharp decline from *Edge Detection* stems from its subtractive nature, causing lower contrast or fine-grained visual anchors, such as small text and thin logos, to

be removed or distorted. Without these key visual cues, models lose the necessary context to form targeted and specific search keywords, causing performance to collapse (Figure 12).

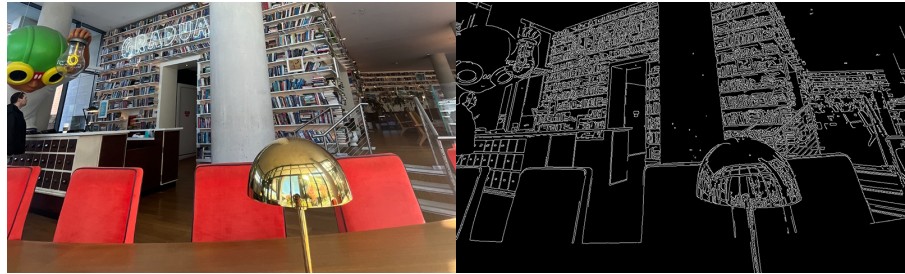

Figure 12: Edge detection ablation example. The accompanying query for this entry reads: *"I was visiting this hotel while attending a conference. Who designed the sculpture to the left?"*. We observe that both color and textual information (e.g., "GRADUA") are completely lost.

**Color ablations alter accuracy and strategy**   Both Black&White (B&W) and Red&Green (R&G) transformations reduce accuracy, reflecting the fact that color frequently serves as discriminative evidence (e.g., differentiating flavors, products, or team jerseys). Color transformations also appear to affect reasoning behavior. We discovered cases where models potentially interpret grayscale inputs as "lower information" and compensate by invoking external tools. In several cases, B&W images led models to use search and find correct answers, whereas color and red-green variants relied on internal guesses or refusals (Figure 13). This suggests that reduced color fidelity may implicitly signal uncertainty and nudge models towards tool-assisted reasoning.

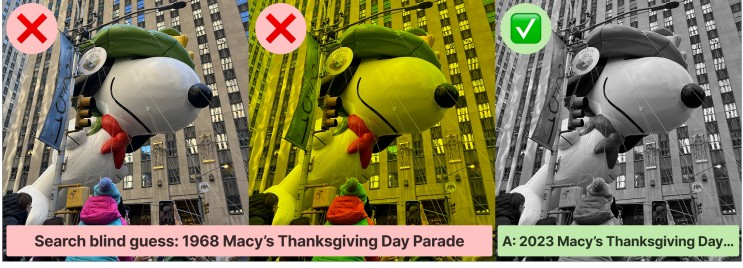

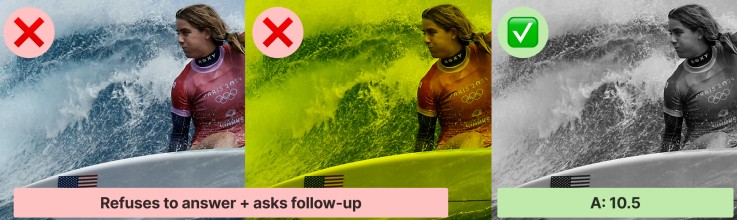

Figure 13: Example entries with color ablations applied. Across both entries, original and R&G similarly ignore tools and fail while the B&W counterpart successfully uses search.

# 7   Conclusion

MFCL Vision provides the first large-scale vision-based benchmark for tool-augmented LLMs, exposes consistent failure modes, and offers lightweight, reproducible metrics that enable rapid iteration. Our analysis reveals that current state-of-the-art models still treat tool use as an optional afterthought, especially under complex, real-world visual conditions, and that simple perturbations such as edge detection can erase nearly all accuracy gains. We release MFCL Vision, all code, and evaluation and analysis tools to spur research on robust, tool-aware reasoning. Future work will

expand MFCL Vision to include additional models and extend beyond the current `search` tool to evaluate multi-tool capabilities on image inputs. This includes tools to manipulate or edit images, as well as paradigms where images are produced as output.

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

# A Appendix

## A.1 Model Error Distribution

Below we provide an error distribution of a sample of leading models run on MFCL Vision. Each model's distribution gives insight into its unique characteristics. For example, we observe that Grok-4 makes greater use of the tool than any other model, but does a poor job of selecting search keywords. Conversely, Claude-Opus-4.1 makes the least use of the tool, yet constructs the most productive search queries.

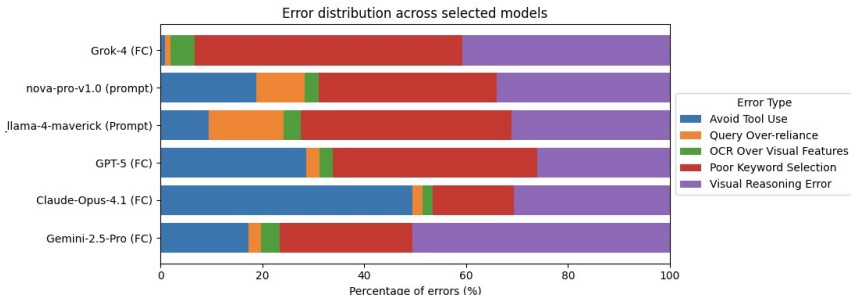

Figure 14: The chart shows the frequency of five error types—avoiding tool use, over-reliance on the provided user query, collapsing into an overly simplistic object character recognition (OCR) task, poor keyword selection, and visual reasoning error—across various models. Error types are color-coded, illustrating differences in model performance.

## A.2 System Prompt for Response Formatting

During evaluation, the model receives explicit instructions for response formatting (via system prompt):

```
 For your final answer to the user, you must respond in this format:
'answer':  A short and precise answer to the question, 'context':  A
brief explanation of how you arrived at this answer or why it is
correct.  If you do not know the answer, respond with:  'answer':  'I
do not know', 'context':  'I do not know'.  If you think the question
cannot be properly answered, respond with:  'answer':  'I cannot answer
this question', 'context':  A short reason explaining why this question
cannot be answered'.
```

## A.3 FC Models vs Prompting Models

Our benchmark evaluates three kinds of information that models can use when forming a search query: (i) text from the provided query, (ii) text found in the image, and (iii) visual features in the image. The way models utilize these sources differs significantly, creating a noticeable gap between prompt-only and FC model approaches. Prompt-only models often ignore image text and visual cues. Instead, they repeat or mimic the provided question in the search call, with no references to visual elements. For example, models searched *"identify building in image square footage"* when asked about a building's size, or *"season in image"* when asked to reason about the season in a busy city scene. These examples demonstrate prompt-only models mistakenly assume the tool can inherently "see" the image, which leads to poor search results.

Within prompt-only models, we also observe differences in how effectively they compensate for this limitation. Some models tend to recognize visual features but use them incorrectly, leading to confident but wrong queries. Other models show better use of image text by weaving it into search queries, however still share the same structural weakness of assuming the tool can "see". In contrast, FC models are explicitly forced to separate and fill arguments for query text, image text, and visual

250  features. This design prevents the "tool sees the image" assumption and leads to more grounded and
251  reliable searches overall.

