# OpenReview forum: "MFCL Vision: Benchmarking Tool Use in Multimodal Large Language Models for Visual Reasoning Tasks"
_NeurIPS.cc/2025/Workshop_Mexico_City/NORA — NeurIPS 2025 Workshop NORA Oral_

### Official Review · Reviewer_4x1d · 2025-11-01
**MFCL Vision: Benchmarking Tool Use in Multimodal Large Language Models for Visual Reasoning Tasks**

**Rating:** 7
**Confidence:** 4

**Review:**

**Overview**

This paper introduces MFCL Vision, a new benchmark designed to evaluate the tool-using capabilities of multimodal large language models in visual contexts. The benchmark comprises 250 tasks, each requiring a model to interpret an image and a textual query to invoke a web-search tool and find a correct answer. The authors also introduce controlled visual perturbations to test model robustness and provide a taxonomy of common failure modes observed in leading models. The work aims to address a gap in existing evaluation frameworks by focusing specifically on the perception-to-tool-use pipeline.

**Weaknesses**

1. The paper states that tasks are "expert-verified" and include "human-produced reasoning traces" but provides little detail on the annotation process itself. The qualifications of the "experts" and metrics for ensuring consistency, such as inter-annotator agreement, are not discussed.

2. Was any manual error analysis conducted to determine the rate of such false negatives? Have you considered supplementing the exact-match score with a semantic similarity metric or human evaluation for a more nuanced assessment?

3. While controlled, the "Edge Detection" ablation is an extreme form of visual degradation that may not be representative of common real-world image corruptions. The practical implications of the performance drop on this specific ablation are therefore less clear.

4. The paper mentions that tasks are solvable by non-expert humans but does not provide a quantitative human performance baseline. This information would be valuable for calibrating the benchmark's difficulty and contextualizing the models' low scores.

5. The error distribution chart in Figure 14 is insightful but lacks normalization. It shows absolute frequencies, making it difficult to compare error tendencies between models that may have attempted different numbers of tasks or failed at different rates.

---

### Official Review · Reviewer_DLve · 2025-11-07
**A Benchmark for Vision-Based Tool Use**

**Rating:** 7
**Confidence:** 4

**Review:**

This paper presents the first large-scale benchmark MFCL Vision designed to evaluate vision-based function calling in MLLMs. The benchmark contains 250 expert-verified tasks that require visual interpretation, formulation of search queries, and correct use of a web-search tool to retrieve answers. It includes tasks across five visual domains and five query types, supports controlled visual perturbations, and defines an automatic exact-match grading scheme. The authors evaluate a wide range of sota models and demo a detailed taxonomy of failure modes affecting MLLM tool use.

Despite the relatively modest dataset size (i.e., 250 tasks), the benchmark is thoughtfully curated, and the evaluation methodology is rigorous. The study’s comparisons across a wide range of foundation models underscore meaningful differences between prompting-based and native function-calling systems. Overall, MFCL Vision fills a critical gap in existing benchmarks and facilitates future work on robust tool use in MLLMs.

---

### Official Review · Reviewer_QtCV · 2025-11-07
**Review for MFCL Vision: Benchmarking Tool Use in Multimodal Large Language Models for Visual Reasoning Tasks**

**Rating:** 7
**Confidence:** 4

**Review:**

The authors introduce MFCL Vision, a large-scale benchmark for vision-based function calling in multimodal large language models (MLLMs). The benchmark dataset consists of 250 tasks across five image domains and five query types. The benchmark computes accuracy based on exact string matching and provides support for ablations and large-scale models. The work's main contribution is establishing a standardized metric for evaluating MLLMs' performance in translating visual inputs into correct tool invocations. Although similar benchmarks exist for text-only tool use or general multimodal understanding, this is the first benchmark of its kind, marking the novelty of the work.

## Pros:
1. The paper provides a novel benchmark dataset. To my knowledge, there are no standard metrics for this specific task.
2. Full feature support: The benchmark provides easy-to-measure metrics based on string matching (as opposed to LLM evaluators) and support for perturbations.
3. The authors present a detailed analysis and taxonomy of common errors (see also Figures 5-11), providing actionable insights for future studies and model development.
4. Strong methodology: The benchmark is well-motivated by similar benchmarks for text tasks and extends an established framework, namely BFCL, for evaluation. An ablation study also revealed interesting limitations and behaviors of the models.

## Cons
1. Dataset size: A dataset of 250 tasks is small for a large-scale benchmark. This could limit the statistical significance of the results across domains and query types and prevent the dataset from capturing the full diversity of real-world scenarios. Text-only benchmarks, in comparison, tend to have thousands or tens of thousands of queries.
2. As the authors acknowledged, the benchmark is limited to a single web search tool. While this is a reasonable simplification for an initial study, it does not reflect the complexity of real-world agentic systems that must select from and orchestrate multiple tools.
3. Task complexity: The tasks are mostly single-step, which does not reflect the complexity of multi-turn, interactive, or complex, multi-step reasoning scenarios for agentic systems. These scenarios are represented by text-only benchmarks.

## Questions
1. If the benchmark dataset is only meant to be used with the string match metric, what is the plan for using the reasoning traces? If it is only used for manual inspection, it seems underutilized for human-provided trusted data.
2. The paper states that all tasks can be solved by a "non-expert human with access to web search." Did you collect human baseline performance with this benchmark? This would help contextualize the model scores.
3. The tuning process involved retaining entries that "consistently defeat state-of-the-art LLMs." Does this curation strategy risk creating an adversarial dataset that highlights primarily model weaknesses rather than representing a distribution of task difficulty more representative of those encountered in the wild?

---

### Official Review · Reviewer_fE5i · 2025-11-07
**Weak accept**

**Rating:** 6
**Confidence:** 3

**Review:**

The authors introduce MFCL Vision, a benchmark for assessing the function-calling capabilities of multimodal LLMs. The capabilities being assessed include how they use various tools (eg. using a search tool) when it comes to a user query. The authors find that the latest state of the art multimodal LLMs avoid using tools, in spite of having access to them, and that they tend to guess the answer based on presented information.




Strengths:




1. The benchmark design is systematic, spanning actress 5 visual domains and 5 query types

2. The evaluation of the models is comprehensive and includes various ablation studies and several multimodal LLMs

3. The taxonomy of the failure modes is novel




Weaknesses:


1. The dataset reproducibility is weak. Though the authors list Google, Reddit, etc. as sources, it does not mention further details such as image licensing, sampling criteria, etc. I also found the annotation details are lacking. No details are given on the number of annotators, if inter-annotator agreement was performed or if tools were used for validation.

2. The evaluation metric might be too strict. Rather than solely relying on exact match accuracy, it might have been useful to see if the scores might have been different if partial matches or semantic-based scoring were considered.

3. The 250 examples do not have the same “difficulty”. I assume some of the scenarios are easier than the other. Hence in this case, it would have been great if the benchmark had a difficulty metric. This could be images where even humans struggled to provide an answer or rated it difficult.

---

### Official Review · Reviewer_MDY1 · 2025-11-07

**Rating:** 6
**Confidence:** 3

**Review:**

The paper proposes a benchmark to test tool use in vision-language models. The authors collect images from the web and develop questions about the images that can only be answered by searching the web for additional context. The study finds that most modern VLMs struggle with the benchmark, where even GPT-5 reaches a maximum accuracy of 34.7% on exact string matching. Failure modes include lack of tool use, bias toward initial search results, and different types of visual reasoning errors. Perturbing the images induces additional failure modes and may also help models in some cases, such as when black and white images are more amenable to visual reasoning.

Strengths:

- The benchmark covers a broad range of vision domains and is grounded in real-world photographs, which will likely improve the utility of the data versus comparable datasets. The rationale for building the dataset are clearly explained (3.1) and are a good fit for agentic systems specifically, e.g., requiring external knowledge via web search.
- The results show that even the most modern models struggle to complete the tasks as expected, which is useful insight for model developers to incorporate into agentic development. The error analysis reveals a wide array of failure modes and is supported by several clear examples, including surprising cases of incorrect model logic (Figures 6, 7). The incorporation of image ablations also yields unexpected findings, and makes one wonder if certain models are trained on ablated data and are therefore more likely to respond correctly to such data. It seems that certain ablations like black-and-white coloring allow the models to focus on more important information (Figure 13), although clearly in aggregate these ablations hurt performance.
- The models tested (Table 1), while mostly not open-source, run the gamut of modern agentic systems and provide options for varying levels of cost/compute for researchers to leverage as needed.

## Weaknesses

- Overall, the dataset is somewhat small compared to similar vision-language datasets, e.g. VQA has data on the order of 10s of thousands. The authors should emphasize that the benchmark is meant more for high-quality evaluation rather than trying to train an agentic system from the ground-up. The limited size of the dataset makes it hard to know how robust the quantitative results are, e.g. if small differences in performance between models are due to genuine differences in model behavior or random noise. Significance testing would help alleviate the concern of dataset size re: consistency of results. Most of the interesting content of the paper is in the qualitative error analysis rather than quantitative results, which is interesting but harder to replicate considering the fuzzy nature of qualitative analysis.
- Important details are missing from the data collection process. It's not clear exactly which websites are used to collect the data (the authors mention "Google, Reddit, Pexels, etc" in Figure 1 but not whether that's the full list), whether those websites' terms of service allow for public data collection, how the authors found photos on the websites (e.g. just random browsing or with specific search terms), whether the prompts are created manually (3.2) or with the iterative help of VLMs (and if so, which VLMs are used), why user personas were included when they are unlikely to change the results of the prompt (e.g. knowing that a user is a food critic in Figure 3 doesn't change the agent's likely course of action), and generally how much human labor the data generation process required (and if significant, how the humans were compensated for their work). Providing as much quantitative detail as possible is important if future work seeks to replicate or extend the data generation process.
- Without an actual human baseline, it's impossible to judge whether the models tested on the benchmark are performing at an adequate level. The authors state that the tasks are designed to be human solvable (3.1) but I have trouble believing that a non-expert person with limited time could look at some of the sample images and arrive at the correct answer for the question, e.g. reading the blurry image in Figure 7 or trying to figure out the location of the library in Figure 12. Ideally, the human user would only have access to the same tools as the models and be required to complete the task in a limited time, albeit longer than models' runtime.
- There is a non-zero risk that some of the tasks in the benchmark could be solved simply by VLMs that had memorized the correct answers, e.g. knowing who won a particular sports match at a given time. It may be useful to include performance from non-agent systems as the lowest-bar estimate of performance.
- In addition to model accuracy (Table 1), it would be useful to report model runtime, tool calls, and/or cost as an extra metric to help show the relative "lift" of using the benchmark. Future users may need to know in advance whether using a more accurate model will also require more tool calls, to anticipate potential cost or compute limitations.
- The authors appear to use exact match in determining accuracy, as mentioned briefly in the introduction (lines 40-41). This is likely to unduly penalize models that return answers in a slightly different format than expected, e.g. omitting articles or reporting numbers in the "wrong" way ("eight" vs. "8"). The authors should consider a more flexible evaluation metric that is less likely to penalize arbitrary variation in response style and would in turn benefit the likely output from a human baseline evaluation (see earlier point).
- It would be useful to report the average % difference between conditions in Table 1 to help understand the overall patterns, e.g. the average % difference between the base condition and each ablation type.

---

### Official Review · Reviewer_aVfZ · 2025-11-07
**Review of "MFCL Vision: Benchmarking Tool Use in Multimodal Large Language Models for Visual Reasoning Tasks"**

**Rating:** 7
**Confidence:** 4

**Review:**

### Summary

- The paper aims to introduce MFCL Vision as the first large-scale benchmark for evaluating vision-based function calling in multimodal LLMs, comprising 250 expert-verified tasks across five image domains and five query types, where models must synthesize visual and textual information to formulate effective web-search queries for answering questions requiring external knowledge.​

- The evaluation methodology exposes models to a singular web-search tool and employs exact-match accuracy on final answers rather than LLM judges, while also examining robustness through controlled visual perturbations including color manipulations, edge detection, and aspect-ratio changes to disentangle perception from reasoning capabilities.​

- The paper aims to present a taxonomy of failure modes organized into three categories: avoiding tool use, poor keyword selection, and visual reasoning errors, with results suggesting that current state-of-the-art models achieve relatively low performance (best model at 34.7% accuracy) and that edge detection ablations produce the largest performance drops.​

### Pros
- The paper addresses a genuine gap by focusing on vision-based function calling at the intersection of visual understanding and tool use, whereas prior benchmarks evaluate either text-only tool use or general multimodal understanding without systematic evaluation of perception-to-tool-use pipelines.​

- The benchmark design demonstrates thoughtful construction principles requiring salient visual hints, external evidence dependence, and image context for disambiguation, ensuring successful completion genuinely requires integration rather than collapsing into pure visual reasoning or guessing.​

- The error taxonomy aims to categorize failure modes systematically across six specific patterns including visual reasoning errors, tool avoidance, poor keyword selection, first-hop bias, over-reliance on query text, and abandoning promising leads, potentially providing actionable diagnostic insights for future model development.​

### Cons
- The benchmark's scale of 250 tasks is modest and tasks are deliberately tuned to "consistently defeat state-of-the-art LLMs," raising concerns about whether it measures genuine capability gaps or primarily captures engineered adversarial examples that may not reflect real-world distribution of vision-based function calling scenarios.​

- The evaluation constrains models to a singular web-search tool, which may not adequately represent real-world agentic scenarios where models must select from multiple tools, coordinate multi-tool workflows, or decide when tool use is unnecessary, potentially limiting ecological validity.​

- The exact-match accuracy metric may be overly rigid and fail to capture semantically equivalent answers or reasonable alternatives, and the paper does not provide detailed analysis of inter-annotator agreement for ground-truth answers or discuss how ambiguity in correct answers was resolved during construction.​

- The paper claims prompt-only models fundamentally assume tools can "see" images, but this conflates model capability with interface design, making it unclear whether observed performance gaps reflect genuine reasoning limitations or simply that the prompting interface is not optimally designed compared to native function calling APIs.​